# Single-Step Genomic Evaluations from Theory to Practice: Using SNP Chips and Sequence Data in BLUPF90

**DOI:** 10.3390/genes11070790

**Published:** 2020-07-14

**Authors:** Daniela Lourenco, Andres Legarra, Shogo Tsuruta, Yutaka Masuda, Ignacio Aguilar, Ignacy Misztal

**Affiliations:** 1Department of Animal and Dairy Science, University of Georgia, Athens, GA 30602, USA; shogo@uga.edu (S.T.); yutaka@uga.edu (Y.M.); ignacy@uga.edu (I.M.); 2Institut National de la Recherche Agronomique, UMR1388 GenPhySE, 31326 Castanet Tolosan, France; andres.legarra@inra.fr; 3Instituto Nacional de Investigación Agropecuaria (INIA), 11500 Montevideo, Uruguay; iaguilar@inia.org.uy

**Keywords:** genomic selection, genomic prediction, genome-wide association, single-step genomic BLUP

## Abstract

Single-step genomic evaluation became a standard procedure in livestock breeding, and the main reason is the ability to combine all pedigree, phenotypes, and genotypes available into one single evaluation, without the need of post-analysis processing. Therefore, the incorporation of data on genotyped and non-genotyped animals in this method is straightforward. Since 2009, two main implementations of single-step were proposed. One is called single-step genomic best linear unbiased prediction (ssGBLUP) and uses single nucleotide polymorphism (SNP) to construct the genomic relationship matrix; the other is the single-step Bayesian regression (ssBR), which is a marker effect model. Under the same assumptions, both models are equivalent. In this review, we focus solely on ssGBLUP. The implementation of ssGBLUP into the BLUPF90 software suite was done in 2009, and since then, several changes were made to make ssGBLUP flexible to any model, number of traits, number of phenotypes, and number of genotyped animals. Single-step GBLUP from the BLUPF90 software suite has been used for genomic evaluations worldwide. In this review, we will show theoretical developments and numerical examples of ssGBLUP using SNP data from regular chips to sequence data.

## 1. Introduction

In the early 1980s, Soller et al. [1] hypothesized that DNA markers like RFLPs (restriction fragment length polymorphisms) would be beneficial in constructing more precise genetic relationships, followed by parentage determination, and the identification of quantitative trait loci (QTL). The high cost of genotyping animals for such markers probably prevented the early widespread use of this technology. When the first draft of the Human Genome Project became available in 2001 [2], one of the most exciting news that came along was that the majority of the genome sequence variation can be attributed to single nucleotide polymorphisms (SNPs). The reality is that SNP markers have become the bread-and-butter of DNA sequence variation [3] and they are now an important tool to determine the genetic potential of livestock. This is because SNPs are abundant, as they are found throughout the entire genome [4], as in introns, exons, promoters, enhancers, or intergenic regions. In fact, there are about three billion nucleotides in the bovine genome, and there are over 30 million SNPs or one every 100 nucleotides is a SNP. Another reason is that SNP genotyping became automatized, relatively cheap, efficient (most loci are read) and highly reproducible (e.g., across laboratories), contrary to microsatellites.

In 2001, Meuwissen et al. [5] envisioned that genomic information could help animal breeders to generate more accurate breeding values, if a dense assay that covers the entire genome become available. Extending the idea of incorporating marker information into best linear unbiased prediction (BLUP), introduced by Fernando et al. [6] and extended to the whole genome by Lande et al. [7] and Haley et al. [8], Meuwissen et al. [5] proposed what is now termed genome-wide selection or genomic selection (GS). The Bayesian models described in Meuwissen et al. [5] provide SNP effects and direct genomic values (DGVs) based on joint analyses of genotypes and phenotypes, method that was easily modified to use pseudo-phenotypes (i.e., estimated breeding values or EBVs adjusted for parent average and accuracy; or progeny deviations) only for genotyped animals as bulls. Following the same line, VanRaden [9] proposed an equivalent method called genomic BLUP (GBLUP), where predictions for genotyped animals are obtained based on genomic relationships (i.e., proportion of alleles shared between animals) instead of pedigree relationships. This genomic relationship matrix is represented by G. After using GBLUP or Bayesian methods, a post-processing step is needed to account for pedigree information; therefore, the traditional BLUP evaluation is still needed. Because several steps are needed to retrieve genomic EBV (GEBV), this class of methods is called multistep. The main advantage of this approach is that the cost is greatly reduced (only selection candidates and highly represented animals such as bulls are genotyped), the traditional BLUP evaluation is kept unchanged and genomic selection can be carried out by using additional analyses. However, the multi-step method has some disadvantages: (a) DGVs are only generated for simple models (i.e., single trait, non-maternal models), which is not the reality of genetic evaluations; (b) only genotyped animals are included in the model; (c) it requires pseudo-phenotypes that are cumbersome to obtain and may rely on accuracy obtained via approximated algorithms [10]. 

Although multistep methods were largely implemented for genomic evaluations worldwide, starting from 2009, this class of methods was not going to be the enduring process to compute genomic predictions. This is because only a fraction of pedigreed animals is genotyped and the genomic information cannot be extended to non-genotyped animals; therefore, genotyped animals have GEBV and non-genotyped have EBV. As a result, several adjustments were proposed, especially in dairy cattle, to make EBV comparable to GEBV under multistep evaluations [11,12], and it was acknowledged that multi-step methods would eventually lead to bias predictions because BLUP predictions would ignore the effects of genomic selection [13]. Intending to solve these problems and to reduce the burden in obtaining genomic predictions, Misztal et al. [14] proposed a method that combines phenotypes, pedigree, and genotypes into a single evaluation. This method is called single-step genomic BLUP (ssGBLUP) and involves replacing the pedigree relationship matrix in the traditional BLUP by a realized relationship matrix, which combines pedigree and genomic relationships. This realized relationship matrix is referred to as the **H** matrix. If the question is why **H**, the answer is quite simple: If the genomic relationship is represented by **G**, just pick the next letter in the alphabet.

Still in 2009, Legarra et al. [15] showed that the pedigree relationship can be viewed as a priori relationship and the genomic relationship as the observed relationship. The derivation of the joint distribution of pedigree and genomic relationships would allow the extension (or imputation) of genomic information to non-genotyped animals. This means that in ssGBLUP pedigree relationships for non-genotyped animals are enhanced by the genomic information of their relatives. Aguilar et al. [16] and Christensen et al. [17] finally showed that although **H** is quite complex, its inverse is rather simple. This development was the landmark for the implementation of ssGBLUP in livestock populations. After 10 years, ssGBLUP has become the preferred tool for genomic evaluation and selection in many livestock species, namely beef cattle [18], pigs [19,20], broilers [21,22], layers [23], dairy sheep and goat [24], meat sheep [25], and fish [26]. Although ssGBLUP adds simplicity to the genomic evaluation system, its implementation involves several details and requires knowledge about peculiarities of the method. In this review, we will show theoretical developments and numerical examples of ssGBLUP, from the BLUPF90 software suite, that will ease the steps toward the application of the method. Although the focus is on BLUPF90, we recognize there are other packages available for computing BLUP-based predictions with and without genomic information. Examples are ASREML [27], Wombat [28], Mix99 [29], DMU [30], MTG2 [31], GCTA [32], among others.

## 2. Software, Methods, and Algorithms 

### 2.1. BLUPF90 Software Suite

BLUPF90 is a collection of software for computations with focus on applications in breeding and genetics. It is based on Fortran 90/95 and started being developed in 1997 by Ignacy Misztal, with the objective to be simple and flexible for model fitting. The first idea was to have a simple BLUP program to compute solutions for the mixed model equations (MME), then blupf90 was the first software created. This software supports general multiple-trait models, different model design per trait, multiple effects, missing data, random correlated and non-correlated effects, dominance effects, and can use several pedigree files or different covariance structures supplied by the user [33]. 

After the first software (i.e., blupf90), several programs were developed to support variance components estimation for linear models (i.e., remlf90, airemlf90, gibbsf90) and linear-threshold models (thrgibbsf90), large-scale genetic evaluations using linear models (blup90iod) and linear-threshold models (cblup90iod), and accuracy approximation (accf90). For information on how to download and use the programs, check Appendix A.

Additionally, a renumbering program (i.e., renumf90) was created that also provides data statistics, performs extensive pedigree checks, can assign unknown parent groups (UPG), supports large data sets, and creates a parameter file that can be used as input for all software in the BLUPF90 suite (see Appendix B). 

When genomic information became available and ssGBLUP was developed, the flexibility of the BLUPF90 family of programs allowed the efficient incorporation of genomics. The extra file with gene content for each animal is easily read, then genomic relationships are computed and can be used by any software in the family. This is because all the programs share the same genomic library, which contains all functions to deal with genomic data. Additionally, software was developed (i.e., pregsf90) to perform quality control and preprocessing of genomic data, and to be the main interface to the genomic library (see Appendix C). All the programs, except the ones for large-scale evaluations, are freely available for research and academic purposes. Linux, Windows, and Mac versions can be downloaded here: http://nce.ads.uga.edu/html/projects/programs. General descriptions about all the programs are available here http://nce.ads.uga.edu/wiki/doku.php?id=application_programs. The current free software can handle up to 25,000 genotyped animals; however, this threshold is frequently raised. Additionally, all software in the BLUPF90 family is under constant development, where the main objective is to improve methods and computing performance. New updates on BLUPF90 are released several times a year.

### 2.2. Genomic Relationship-Based Methods

Single-step GBLUP is considered a genomic relationship-based method. This class of methods use SNPs to infer relationships among individuals, quantifying the number of alleles shared between two individuals. Genomic relationships are identical by state (IBS) because they account for the probability that two alleles randomly picked from each individual are identical, independently of origin. Pedigree relationships are identical by descent (IBD) because they consider that the shared alleles come from the same ancestor in a base population. 

Now we detail the ideas from VanRaden [9]. Assuming a matrix of SNPs inherited by each animal (M), with dimension *n* × *m* where *n* is the number of animals and *m* the number of SNPs. Several parametrizations exist, but if AA = 0, AB = 1, and BB = 2, M has to be centered by allele frequency. Assuming a vector p with elements equal to *p_i_*, the frequency of allele B at locus *i*:(1)$$Z=M-2p^{\prime }$$

To understand why Z is a centered matrix of allele content, we can use only one biallelic marker. If the effect of each copy of the A allele is *a* and the frequency of AA is *p*^2^, individuals with AA have a (non-centered) breeding value u = *2a*; individuals aa have u = *0* with a frequency of *q^2^*; individuals Aa have u = *a* with a frequency *2pq*. The variance explained by this marker is $\mathrm{Var}\left(u\right)=E\left(u^{2}\right)-E{\left(u\right)}^{2}$ [34]. The average of *u* is *2ap*^2^ + *a2pq*; which becomes *2pa*. The variance explained by one marker is: *(2a)*^2^*p*^2^ + *2pq(a)*^2^ − (*2pa*)^2^ = *2pqa*^2^. Given the average of u is *2pa*, as shown above, we can compute the covariance between individuals *i* and *j* for this marker. If we express the breeding values of the animals *i* and *j* as *ma* deviated from the population mean [34], we obtain Equations (2) and (3): (2)$$u_{i}=\;m_{i}a-2p^{\prime }a=\;(m_{i}-2p^{\prime })a=z_{i}a$$
(3)$$u_{j}=\;m_{j}a-2p^{\prime }a=(m_{j}-2p^{\prime })a=z_{j}a$$

According to Legarra et al. [34], if Var(a)=
$I\sigma_{a}^{2}$, or marker variance, and the genetic variance in Hardy–Weinberg equilibrium is $2\sum p_{i}q_{i}\sigma_{a}^{2}$, the rules of variances and covariances can be applied: (4)$$\mathrm{Cov}\left(u_{i},u_{j}\right)=(z_{i}-2p)a(z_{j}-2p)a=(z_{i}-2p)(z_{j}-2p)\sigma_{a}^{2}$$

If instead of using the allele coding 0,1,2 we use −1,0,1:(5)$$\mathrm{Cov}\left(u_{i},u_{j}\right)=z_{i}z_{j}\sigma_{a}^{2}$$

Dividing the covariance by the genetic variance $2\sum p_{i}q_{i}\sigma_{a}^{2}$, we get realized relationships.

Going from one to several markers, the breeding value of an animal can be calculated as the sum of SNP effects weighted by the genotype content (u=Za). Assuming the same variance per locus, the variance of u is:(6)Var(u)= Var(Za)
(7)$$\mathrm{Var}\left(u\right)=\;Z\;\mathrm{Var}\left(a\right)\;Z^{\prime }$$
(8)$$\mathrm{Var}\left(u\right)=ZZ^{\prime }\sigma_{a}^{2}$$

If the genetic variance $\sigma_{u}^{2}=2\sum_{i=1}^{SNP}p_{i}\left(1-p_{i}\right)\sigma_{a}^{2}$, then $\sigma_{a}^{2}=\frac{\sigma_{u}^{2}}{2\sum_{i=1}^{SNP}p_{i}\left(1-p_{i}\right)}$. Replacing $\sigma_{a}^{2}$ in (8) we have that:(9)$$\mathrm{Var}\left(u\right)=ZZ^{\prime }\frac{\sigma_{u}^{2}}{2\sum_{i=1}^{SNP}p_{i}\left(1-p_{i}\right)}$$
(10)$$\mathrm{Var}\left(u\right)=\frac{ZZ^{\prime }}{2\sum_{i=1}^{SNP}p_{i}\left(1-p_{i}\right)}\sigma_{u}^{2}$$

Therefore, and according to VanRaden [9], the genomic relationship (G) is given by: (11)$$G=\frac{ZZ^{\prime }}{2\sum p_{i}\left(1-p_{i}\right)}$$
then,
(12)$$\mathrm{Var}\left(u\right)=G\sigma_{u}^{2}$$

Therefore, genomic relationships are standardized covariances. When $ZZ^{\prime }$ is divided by $2\sum p_{i}\left(1-p_{i}\right)$, G becomes analogous to the pedigree relationship matrix (A). The G matrix contains the number of homozygous loci for each individual in the diagonals, and the number of alleles shared among individuals in the off-diagonals. Other ways to construct the genomic relationship matrix are described in the literature. For more details, check Leutenegger et al. [35] and Amin et al. [36].

If G is centered using observed allele frequencies, the average over all elements is zero and average diagonal is 1 when there is no inbreeding. However, it is only when base allele frequencies are used that elements of G can be interpreted as elements of A (this will be more detailed later). In general, G traces inbreeding much further than A because of its IBS nature and because A is limited by the recent pedigree recording. 

When the number of genotyped animals is bigger than the number of SNPs, or if there are similar individuals (e.g., clones), G becomes singular; therefore, cannot be inverted. To overcome this problem, usually, G is “blended” with a small percentage of an identity matrix or the pedigree relationship matrix among genotyped animals ($A_{22}$):(13)$$G=\alpha \;G+(1-\alpha )\;A_{22}$$
where the blending parameter α is usually 95% but can vary from 99 to 80% [37], or even to 50% [38]. The blending parameter (1 − *α*) can be understood as the fraction of genetic variance not explained by markers and computed by maximum likelihood methods (see below).

### 2.3. From GBLUP to ssGBLUP

Understanding the difference between GBLUP and ssGBLUP is a crucial step. Because there is still a lot of confusion, an explanation about GBLUP is provided. 

The GBLUP is equivalent to SNP-BLUP, but in GBLUP genomic breeding values (u=Za) are estimated, instead of SNP effects (a) in SNP-BLUP. It also assumes that SNPs have a priori a normal distribution; the majority of SNPs have a small effect, and very few have moderate to large effect. Using a simple animal model as shown in (14) and (15):(14)y= Xb+Wu+e
(15)$$\left[\begin{array}{cc} X^{\prime }X & X^{\prime }W\\ W^{\prime }X & W^{\prime }W+G^{-1}\frac{\sigma_{e}^{2}}{\sigma_{u}^{2}} \end{array}\right]\left[\begin{array}{c} \hat{b}\\ \hat{u} \end{array}\right]=\left[\begin{array}{c} X^{\prime }y\\ W^{\prime }y \end{array}\right]$$
where W is the incidence matrix for animal effect (u), X is the incidence matrix for fixed effects (b), $\sigma_{e}^{2}$ is the residual variance, and $u\sim N\left(0,G\sigma_{u}^{2}\right)$. 

Therefore, GBLUP is a BLUP where A is replaced by the genomic relationship matrix. The effectiveness of GBLUP will depend on the ability of G to approach the realized genetic relationships. In addition, performing a quality control of genomic data before constructing G avoids biases and losses of accuracy.

If we assume that not all the genetic variance is explained by markers, an extra polygenic effect can be included to explain the remaining variance. In this case, the model in (14) becomes:(16)y= Xb+Wu+Wg+e
where g is a vector of residual polygenic effect that is not captured by the SNPs. Assuming that α is the proportion of variance explained by SNPs, the total additive genetic effect ($u_{g}$) becomes
(17)$$u_{g}=u+g$$
(18)$$\mathrm{Var}(u_{g})=\alpha G\sigma_{g}^{2}+\left(1-\alpha \right)A_{22}\;\sigma_{u}^{2}$$

Therefore,
(19)$$G=\alpha G+\left(1-\alpha \right)A_{22}$$

In real situations, it is assumed that α varies from 0.8 to 0.95. Note that this is also going to make G invertible [17]. When (1−α) is used strictly to make G (semi-) positive definite, it is called a blending parameter.

Although GBLUP has been widely used in animal and plant breeding applications, its main problem is that only genotyped animals are in the model. As only a fraction of animals is genotyped, GBLUP may have less phenotypic and pedigree information than BLUP. Because of that, some extra steps are needed to combine genomic and pedigree information. When using GBLUP, SNP-BLUP or Bayesian models, the genomic evaluation method is called multistep. The steps involved in multistep are: (1) Estimation of EBV using traditional BLUP (i.e., all available information); (2) de-regression of EBV, which condenses information from phenotypes (e.g., daughter yield deviation in dairy cattle); (3) estimation of SNP effects using GBLUP or other models; (4) prediction of Za, which is also known as direct genomic values (DGVs); (5) blending DGVs with average of parent’s EBV, which is known as parent average (PA), with published EBV, or with PTA. The main issue on having an evaluation with several steps is that some errors and biases can be introduced during those steps [10], and that BLUP will not be robust to genomic selection decisions [13]. 

The idea for ssGBLUP came from the fact that usually only a small portion of the animals, in a given population, is genotyped. In this way, the best approach to avoid several steps would be to combine pedigree and genomic relationships and use this matrix as the covariance structure in the MME. Legarra et al. [15] stated that genomic evaluations would be simpler if genomic relationships were available for all animals in the model. Then, their idea was to look at A as a priori relationship and to G as observed relationships; however, G is observed only for some individuals, and those individuals have $A_{22}$ as a priori relationship. Based on that, it was shown that the genomic information could be extended to non-genotyped animal based on the joint distribution of breeding values of non-genotyped ($u_{1}$) and genotyped ($u_{2}$) animals [15,17]:(20)$$p\left(u_{1},u_{2}\right)=\;p\left(u_{2}\right)p(u_{1}|u_{2})$$
(21)$$p\left(u_{2}\right)=\;N\left(0,G\right)$$

If we consider that
(22)$$\mathrm{Var}\left(u\right)=A\sigma_{u}^{2}$$
(23)$$A=\left[\begin{array}{cc} A_{11} & A_{12}\\ A_{21} & A_{22} \end{array}\right]$$
where subscripts 1 and 2 represent non-genotyped and genotyped animals, respectively. The conditional distribution of breeding values for non-genotyped and genotyped animals is
(24)$$p\left(u_{1}|u_{2}\right)=\;N\left(A_{12}A_{22}^{-1}u_{2},A_{11}-A_{12}A_{22}^{-1}A_{21}\right)$$

If $u_{2}$ in $A_{12}A_{22}^{-1}u_{2}$ is replaced by a vector of observed gene content, the formula can be used to estimated gene content for non-genotyped animals based on observed gene content for genotyped animals [39]. It implies that by using $A_{12}A_{22}^{-1}u_{2}$ the genomic information can be implicitly imputed from genotyped animals to non-genotyped based on pedigree relationships. The variance in the distribution ($A_{11}-A_{12}A_{22}^{-1}A_{21}$) is the prediction error term. Therefore, because the animals with subscript 1 have no genotypes, the variance depends on their pedigree relationships with genotyped animals. In this way, variances and covariances are:(25)$$\mathrm{Var}\left(u_{1}\right)=\mathrm{Var}\left(A_{12}A_{22}^{-1}u_{2}+\varepsilon \right)\;=\mathrm{Var}(A_{12}A_{22}^{-1}u_{2})+\mathrm{Var}\left(\varepsilon \right)\;=A_{12}A_{22}^{-1}GA_{22}^{-1}A_{21}+A_{11}-A_{12}A_{22}^{-1}A_{21}$$

Rearranging:
$$=A_{11}+A_{12}A_{22}^{-1}GA_{22}^{-1}A_{21}-A_{12}A_{22}^{-1}A_{21}\;=A_{11}+A_{12}A_{22}^{-1}GA_{22}^{-1}A_{21}-A_{12}A_{22}^{-1}IA_{21}\;\mathrm{Var}\left(u_{1}\right)=A_{11}+A_{12}A_{22}^{-1}GA_{22}^{-1}A_{21}-A_{12}A_{22}^{-1}A_{22}A_{22}^{-1}A_{21}$$

Therefore,
(26)$$\mathrm{Var}\left(u_{1}\right)=A_{11}+A_{12}A_{22}^{-1}\left(G-A_{22}\right)A_{22}^{-1}A_{21}$$
(27)$$\mathrm{Var}\left(u_{2}\right)=\mathrm{Var}\left(Za\right)=G$$
(28)$$\mathrm{Cov}\left(u_{1},u_{2}\right)=\mathrm{Cov}\left(A_{12}A_{22}^{-1}u_{2},u_{2}\right)=A_{12}A_{22}^{-1}\mathrm{Var}\left(u_{2}\right)$$
(29)$$\mathrm{Cov}\left(u_{1},u_{2}\right)=A_{12}A_{22}^{-1}G$$

Finally, the matrix that contains the joint relationships of genotyped and non-genotyped animals is given by:(30)$$H=\left(\begin{array}{cc} \mathrm{Var}\left(u_{1}\right) & \mathrm{Cov}\left(u_{1},u_{2}\right)\\ \mathrm{Cov}\left(u_{2},u_{1}\right) & \mathrm{Var}\left(u_{2}\right) \end{array}\right)=\left(\begin{array}{cc} A_{11}+A_{12}A_{22}^{-1}\left(G-A_{22}\right)A_{22}^{-1}A_{21} & \;A_{12}A_{22}^{-1}G\\ GA_{22}^{-1}A_{21} & G \end{array}\right)$$
(31)$$H=A+\left[\begin{array}{cc} A_{12}A_{22}^{-1}\left(G-A_{22}\right)A_{22}^{-1}A_{21} & \;A_{12}A_{22}^{-1}\left(G-A_{22}\right)\\ \left(G-A_{22}\right)A_{22}^{-1}A_{21} & G-A_{22} \end{array}\right]$$
which can be simplified to:(32)$$H=A+\left[\begin{array}{cc} A_{12}A_{22}^{-1} & \;0\\ 0 & I \end{array}\right]\left[\begin{array}{c} I\\ I \end{array}\right]\left[G-A_{22}\right]\left[\begin{array}{cc} I & I \end{array}\right]\left[\begin{array}{cc} A_{22}^{-1}A_{21} & \;0\\ 0 & I \end{array}\right]$$

This H matrix is; therefore, a relationship matrix constructed with SNP markers and pedigree, where the SNP information is projected to the individuals that are not genotyped. Some of its properties include being always semi-positive definite and being positive definite and invertible if G is invertible. Although H is very complicated, its inverse ($H^{-1}$) is quite simple [16,17]: (33)$$H^{-1}{=A}^{-1}+\left[\begin{array}{cc} 0 & 0\\ 0 & G^{-1}-A_{22}^{-1} \end{array}\right]$$

As both $A^{-1}$ and $G^{-1}$ capture relationships, $A_{22}^{-1}$ should be subtracted to avoid double-counting of pedigree information for genotyped animals. 

Assuming the following animal model:(34)y=Xb+Wu+e

The MME for ssGBLUP becomes:(35)$$\left[\begin{array}{cc} X^{\prime }X & X^{\prime }W\\ W^{\prime }X & W^{\prime }W+H^{-1}\frac{\sigma_{e}^{2}}{\sigma_{u}^{2}} \end{array}\right]\left[\begin{array}{c} \hat{b}\\ \hat{u} \end{array}\right]=\left[\begin{array}{c} X^{\prime }y\\ W^{\prime }y \end{array}\right]$$

The distribution of u becomes:(36)$$u\sim N\left(0,H\sigma_{u}^{2}\right)$$

Therefore, the only difference between BLUP and ssGBLUP is that $A^{-1}$ is replaced by $H^{-1}$. Subsequently, all tools based on BLUP mixed model equations, as the restricted maximum likelihood (REML [40]), can be easily converted to single-step. In a nutshell, if all animals are genotyped, ssGBLUP becomes GBLUP, but if no animals are genotyped, ssGBLUP becomes BLUP.

Advantages of ssGBLUP include simplicity of use, simultaneous fit of genomic information and estimation of fixed effects [10], relatively higher accuracy than multistep methods [41,42,43,44,45], potential to account for pre-selection bias as all pedigree, phenotypic, and genomic information can be included in the model [12,13], and can be easily extended to any model.

### 2.4. Applying ssGBLUP to a Simulated Data Using blupf90

A dataset that mimicked a cattle population was simulated using QMSim [46]. Pedigree information and phenotypes for 10,000 animals, and genotypes for 1020 parents from generations 1–4 and 1004 individuals in generation 5 were generated. Files with pedigree, phenotypes, and genotypes are available at https://github.com/danielall/Data_ssGBLUP. Shortly, the pedigree file is named pedigree.txt and contains three columns: animal, sire, and dam. The phenotype file is named phenotypes.txt and contains animal, sex, phenotype, true breeding value, and generation. Phenotypes (y) were generated as y = sex_effect + true_breeding_value + residual. Genotypes were coded based on the number of copies of the alternative allele (0, 1, 2) and are in a file named genotypes.txt, with: animal and SNP_genotype. The last file (gen_map.txt) contains the map for SNPs: SNP identification, chromosome number, position (in base pairs).

After running renumf90 to renumber the data (see Appendix B), the renumbered phenotype file is named renf90.dat and contains phenotype, renumbered sex code, and renumbered animal ID; the renumbered pedigree file is renadd02.ped; and the parameter file generated by renumf90 is named renf90.par (Box 1). This parameter file was created based on the following model: y = sex + u + residual, where u is the animal effect or direct additive genetic effect. To run ssGBLUP, blupf90 can be used with the parameter file given in Box 1 (see Appendix B for a description of keywords and values). The following command line can be used to save the screen output to a file: 

blupf90 renf90.par | tee blupout.log

The above command will provide the parameter file when blupf90 asks for it and will save the screen output to a file named blupout.log.

Box 1Parameter file for running ssGBLUP in blupf90.
DATAFILE
 renf90.dat
NUMBER_OF_TRAITS
   1
NUMBER_OF_EFFECTS
   2
OBSERVATION(S)
 1
WEIGHT(S)

EFFECTS: POSITIONS_IN_DATAFILE NUMBER_OF_LEVELS TYPE_OF_EFFECT[EFFECT NESTED]
 2  2 cross 3 12010 cross
RANDOM_RESIDUAL VALUES
  0.60000 RANDOM_GROUP 2 RANDOM_TYPE add_an_upginb FILErenadd02.ped
(CO)VARIANCES
  0.40000OPTION SNP_file genotypes.txtOPTION map_file gen_map.txt

Preconditioner conjugate gradient [47] is the default method used by blupf90 to solve the MME; however, other options exist. To check all options blupf90 can take, check this link: http://nce.ads.uga.edu/wiki/doku.php?id=readme.blupf90.

The output file provided by blupf90 with solutions for all effects is the “solutions” file, and the first 5 lines of this file are shown in Box 2. The first line is a header indicating columns for trait, effect, level, and solution. In this example, only one trait was used, so all entries in the trait column are 1; the effect column contains the number of the effects in the model (i.e., sex and animal effect); level refers to the levels of the effects (i.e., 2 for sex and 12,010 for animal effect (direct additive genetic)); the last column contains the solutions for all levels of the effects in the model. As ssGBLUP was used by blupf90 because the option OPTION SNP_file was included, solutions of the animal effect are GEBV for both genotyped and non-genotyped animals. It is important to remember the effects were renumbered using renumf90, so the original and renumbered levels for fixed effects and animal effect are in renf90.tables and renadd02.ped, respectively (see Appendix B). 

Box 2First five lines of the blupf90solutions file.trait/effect level  solution 1 1   1    2.43346240 1 1   2    1.44508009  1 2   1    0.05317279 1 2   2    −0.05317279 

To have GEBV matched back to the original ID, a simple R script, as the one in Box 3, can be used. 

Box 3Merging GEBV with original animal ID in R.
rm(list=ls())

sol<-read.table(“solutions”, skip=1)

sol_gebv<-subset(sol,sol[,2]==2)

names(sol_gebv)<-list(“trait”,“effect”,“level”,“solutions”)

ped<-read.table(“renadd02.ped”)

ids<-data.frame(ped[,1],ped[,10])

names(ids)<-list(“level”,“orig_level”)

sol_orig_id<-merge(ids,sol_gebv,by=“level”)

write.table(sol_orig_id,file=“sol_orig_id.txt”,quote=F,row.names=F)


The blupf90 software outputs a large amount of information on the screen, including quality control checks, statistics for G and $A_{22}$ and respective inverses, and statistics for $G^{-1}-A_{22}^{-1}$. This is because when the option OPTION SNP_file is used in blupf90, it turns the genomic library on and all checks are done. To avoid doing quality control of genomic data when using blupf90, add the following option at the end of the parameter file: OPTION no_quality_control. The genomic library has an interface software called preGSf90, which contains a myriad of options. To check all options available in the genomic library: http://nce.ads.uga.edu/wiki/doku.php?id=readme.pregsf90. To see how to use preGSf90 to perform quality control and preprocessing of genomic data, check Appendix C.

### 2.5. Compatibility between Pedigree and Genomic Relationships

Based on how H is constructed, the central element is $G-A_{22}\;$(see Equation (31)), which implies both matrices should be compatible [10,48]. Compatibility can be understood as both matrices referring to the same genetic base and to the same genetic variance. However, genomic relationships can be biased if G is constructed based on allele frequencies other than the ones from the base population [9]. However, allele frequencies from the base population are not known because of the recent recording of pedigrees (i.e., the base population *per se* is unknown). Although those frequencies can be estimated using the method proposed by Gengler et al. [39], these estimates are not very accurate because the base population is several generations away from the genotyped individuals. Additionally, in certain contexts such as missing pedigrees there is not a uniquely defined base population. Most commonly, allele frequencies based on the recent genotyped population are used to construct G. When this is the case, the expectation of breeding values for genotyped animals is 0 [9]. However, if the population is under selection, mean breeding values should change from the base population to the genotyped individuals (i.e., they should deviate from 0). To account for selection and for the fact genotyped animals are more related through $A_{22}$ than G is able to reflect (i.e., especially when current allele frequencies are used), Vitezica et al. [48] proposed an adjustment factor (ρ) to match averages of G to averages of $A_{22}$. This adjustment was crucial to avoid bias in ssGBLUP evaluations, especially in populations under selection. It can be calculated as:(37)$$\rho =\frac{1}{n^{2}}(\sum_{i}\sum_{j}A_{22\;i,j}\;-\sum_{i}\sum_{j}G_{i,j})$$
where n is the number of elements in $A_{22}$ and G. The new G is constructed as
(38)$$G*=(1-\;\rho /2)\;G+11^{\prime }\;\rho$$

G* is the adjusted genomic relationship matrix, 1 is a vector of ones, and ρ is Wright’s F_ST_, which models the difference between pedigree and genomic base by implicitly fitting a constant μ, unlike in Hsu et al. [49] where the constant is fit explicitly.

When ssGBLUP was first implemented [16] in the BLUPF90 family of programs, $A^{-1}$ was constructed based on Henderson [50] and Quaas [51] assuming no inbreeding, a frequently-used approximation [52,53], $G^{-1}$ was constructed based on VanRaden [9], and $A_{22}^{-1}$ was based on Colleau [54] and fully considered inbreeding. As the algorithms to construct $G^{-1}$ and $A_{22}^{-1}$ implicitly consider inbreeding, but not the algorithm for $A^{-1}$, $H^{-1}$ was often ill-conditioned because of the unbalance between $A^{22}$ (i.e., the portion of $A^{-1}$ for genotyped animals) and $A_{22}^{-1}$, which has larger coefficients due to inbreeding. This would lead to convergence problems and overestimation of GEBV. To solve this problem, scaling factors to decrease the amount of information in $A_{22}^{-1}$ (ω) and to increase in $G^{-1}$ (τ) were proposed [16,55]:(39)$$H^{-1}{=A}^{-1}+\left[\begin{array}{cc} 0 & 0\\ 0 & \tau G^{-1}-\;\omega A_{22}^{-1} \end{array}\right]$$

Primarily, ω controls inflation due to incompleteness of pedigree and τ controls additive genetic variance [56]. The ω parameter was usually set to 0.7 for beef and dairy cattle ssGBLUP evaluations, and from 0.5 to 0.8 for pig evaluations. The appropriate value depended on the reduction of overestimation, which was evaluated based on validation studies. However, in 2016 the BLUPF90 developers observed that when inbreeding was considered in $A^{-1}$ by adding an extra option to the renumf90 parameter file (see below), the need for ω lower than 1 was reduced. It was rather surprising that ignoring inbreeding in the set-up of $A^{-1}$, which is harmless in BLUP applications, had such a great impact in ssGBLUP. In fact, when genotyped animals have complete pedigree, τ and ω are likely to be equal to 1. Therefore, the compatibility among $A^{-1}$, $G^{-1}$, and $A_{22}^{-1}$ is the key to avoid the use of ad-hoc scaling parameters while keeping GEBV with an acceptable level of inflation/deflation. To ensure consideration of inbreeding in the set-up of $A^{-1}$ the lines


INBREEDING

pedigree


need to be included in the parameter file for renumf90, and then the genetic effect in the parameter file for blupf90 needs to be
RANDOM_TYPEadd_an_upginb

### 2.6. Changing Blending, Tuning, and Scaling Parameters in blupf90

By default, in the blupf90 the blending parameter α is set to 0.95, which makes 1−α (or β) equal to 0.05. This is used to overcome singularity problems (i.e., G being non-positive definite). Using lower values for α can speed up convergence, with small or no impact on accuracy. To change α and β in blupf90, assuming the new values would be 0.90 and 0.10, the following option can be added to renf90.par:OPTION AlphaBeta 0.90 0.10

To model the difference between pedigree and genomic base, which is very important to reduce bias in GEBV, the default in the genomic library is to adjust G as proposed in Chen et al. [57]: $G^{*}=\varphi G+\delta$, where $\varphi =\left[\frac{\bar{dıagA_{22}}-\bar{offdıagA_{22}}}{\bar{dıagG}-\bar{offdıagG}}\right]$ and $\delta =\bar{dıagA_{22}}-\bar{dıagG}*\varphi$. To change the adjustment of G to the one proposed by Vitezica et al. [48] and demonstrated in Equation (38), the following option can be added to the blupf90 parameter file (e.g., renf90.par):OPTION tunedG 4

A total of four different adjustments are implemented in the BLUPF90 family of programs; however, types 2 [57] and 4 [48] are more frequently used. To see other options, check this link: http://nce.ads.uga.edu/wiki/doku.php?id=readme.pregsf90.

If GEBV are underestimated/overestimated, ad-hoc scaling factors can be used to control the amount of information in $A_{22}^{-1}$ (ω) and in $G^{-1}$ (τ). The default in blupf90 is ω = τ = 1. To change those values, an option can be added to the blupf90 parameter file. Supposing only ω is to be changed to 0.95, whereas τ is still 1:OPTION TauOmega 1.0 0.95

Values of ω smaller than 1 helps to avoid overestimation; however, caution is recommended when using this option. A careful investigation of coefficients of the regression of a benchmark variable on GEBV in cross-validation studies is recommended when the objective is to choose an appropriate value.

### 2.7. Estimating SNP Effects in ssGBLUP

Even though ssGBLUP is a genomic relationship-based method and provides GEBV as final output, SNP effects can still be calculated in this method. This is because GBLUP is equivalent to SNP-BLUP [9] as u=Za and Var(u)=Var(Za). Using this idea, the selection index equation for GBLUP can be represented by:(40)$$\hat{u}=G{\left[G+R\left(\frac{\sigma_{a}^{2}}{\sigma_{e}^{2}}\right)\right]}^{-1}\left(y-X\hat{b}\right)$$
where R is a diagonal matrix with (heterogeneous if needed) residual variance. If $\hat{u}|\hat{a}=Z\hat{a}$, replacing the first G by $Z^{\prime }$, weighted by the ratio of SNP to additive direct variances (i.e., k = $\sigma_{a}^{2}/\sigma_{u}^{2}$), would allow the calculation of SNP effects (a) [9]:(41)$$\hat{a}=Z^{\prime }k{\left[G+R\left(\frac{\sigma_{a}^{2}}{\sigma_{e}^{2}}\right)\right]}^{-1}\left(y-X\hat{b}\right)$$

As we saw before, $\sigma_{a}^{2}=\sigma_{u}^{2}/2\sum p_{i}\left(1-p_{i}\right)$. Therefore, k can be reduced to $1/2\sum p_{i}\left(1-p_{i}\right)$. Assuming that: (42)$$w={\left[G+R\left(\frac{\sigma_{u}^{2}}{\sigma_{e}^{2}}\right)\right]}^{-1}\left(y-X\hat{b}\right)$$
then,
(43)$$\hat{a}=kZ^{\prime }w$$
therefore,
(44)$$\hat{u}=Gw$$

In this way,
(45)$$w=G^{-1}\hat{u}$$

Finally, the SNP effects can be calculated as in (46):(46)$$\hat{a}=kZ^{\prime }G^{-1}\hat{u}$$
as Var(a)=D, a diagonal matrix of SNP variances, the conditional mean of SNP effects given the GEBV is:(47)$$\hat{a}|\hat{u}=k{DZ}^{\prime }G^{-1}\hat{u}$$

Thus, given GEBV from ssGBLUP are available, SNP effects are calculated as [58]:(48)$$\hat{a}=k{DZ}^{\prime }G^{-1}\hat{u}$$

If SNP effects are available, indirect predictions (IP) can be calculated for young genotyped animals in between official ssGBLUP evaluations, as the sum of SNP effects weighted by gene content [18]. Except for the small portion of remaining pedigree variation, they are identical to GEBV [18]. Indirect predictions may also be useful for genotyped animals that have incomplete pedigree. Such animals can increase bias and reduce reliability of GEBV if included in official ssGBLUP evaluations, given that A is poorly constructed for them and results in incompatibilities with G (which is correct) [56]. Additionally, if lots of animals are genotyped, say weekly, but they do not contribute to the evaluation (as in young animals that do not have phenotype or progeny yet), having IP for them reduces computing costs. 

Another feature of having SNP effects is the ability to account for the fact that SNPs explain different proportion of genetic variance on the trait, and this leads to iterative methods similar to Bayesian regressions (i.e., BayesA, BayesB). An iterative method was proposed to add different weights for SNPs under ssGBLUP, which is called weighted ssGBLUP [58]. In this method, seven steps are needed:

Set the diagonal matrix of SNP variance or weight as an identity, **D = I**Compute the genomic relationships: $G={ZDZ}^{\prime }/k$, where $k=1/2\sum p_{i}\left(1-p_{i}\right)$Run ssGBLUP to obtain $\hat{u}$Convert $\hat{u}$ into SNP effects: $\hat{a}=k{DZ}^{\prime }G^{-1}\hat{u}$Estimate SNP variance for SNP *i* e.g., as $d_{i}=a_{i}^{2}$ (i.e., quadratic weight)Normalize DIterate from 2 until changes in SNP variance are small across iterations

Usually, the best weights are obtained after one to two rounds. Different formulas can be used to calculate SNP variance, but all of them are approximations. Several authors have reported decrease in GEBV accuracy and increase in bias over iterations [59,60] when variance is calculated based on squared SNP effects, especially for more polygenic traits. This is because SNP variance would reach extreme values over iterations. VanRaden [9] proposed and successfully applied in US dairy cattle a formula to calculate SNP variance that limits the change over iterations, avoiding extreme values. This method is called non-linearA:(49)$$d_{i}={\mathrm{CT}}^{\frac{\left|{\hat{a}}_{i}\right|}{\sigma \left(\hat{a}\right)}-2}$$
where CT is a constant that determines the departure from normality; $\left|{\hat{a}}_{i}\right|$ is the absolute estimated SNP effect for marker i, and $\sigma \left(\hat{a}\right)$ is the standard deviation of the vector of estimated SNP effects. Garcia et al. [26] and Fragomeni et al. [61] showed that non-linearA had good convergence properties and avoided extreme values. The maximum change in variance is usually limited by the minimum between 5 and the exponent of CT; whereas CT was empirically derived as 1.125 over several polygenic traits for dairy cattle populations [9], meaning the distribution for SNP effects approaches a *t* distribution with large degrees of freedom, e.g., approaching a normal distribution.

Considering SNP variances when constructing G in ssGBLUP seems to improve the accuracy of predicting GEBV for data sets with small number of genotyped animals, but marginal or no improvement was observed for large genotyped populations (i.e., >10 k genotyped animals) [60], even for less polygenic traits. If the data allows to accurately estimate SNP effects, there is no advantage in selecting SNPs and tagging chromosome segments differently. The fact that SNP selection does not improve accuracy with large datasets benefits commercial evaluations that use multiple-trait models, as models with different SNPs per trait are easy to implement for single- but not multiple-trait models [62].

Once the variance for each SNP is calculated, the proportion of additive genetic variance can be plotted for all SNPs in a Manhattan plot. A threshold of 1% of genetic variance can be assumed if the objective is to explore associations between traits and regions in the genome, like in genome-wide association studies (GWAS). 

More formally, and according to the common use in ambitious GWAS studies, *p*-values for SNPs can be calculated as [63,64,65]:(50)$$pval_{i}=2\left(1-\Phi \left(\left|\frac{{\hat{a}}_{i}}{sd\left({\hat{a}}_{i}\right)}\right|\right)\right)$$
where Φ is the cumulative standard normal function and $sd\left({\hat{a}}_{i}\right)$ is the square root of prediction error variance (PEV) of the *i*-th SNP effect. Prediction error variance for each SNP effect can be calculated as [65]:(51)$$var\left({\hat{a}}_{i}\right)=k\;\alpha \;b\;z_{i}^{\prime }G^{-1}(G\sigma_{u}^{2}-C^{u_{2}u_{2}})G^{-1}z_{i}\;b\;\alpha \;k$$
where $C^{u_{2}u_{2}}$ is the portion of the inverse of the LHS of MME for ssGBLUP (34) referent to genotyped animals; *b* is (1- ρ/2), which is a function of the tuning parameter in (38); k and α were defined previously.

### 2.8. Using postGSf90 to Compute SNP Effects, SNP Variances, and p-Values

If the objective is to backsolve GEBV to SNP effect and then calculate variance explained by SNPs, postGSf90 can be used. This software was primarily developed to serve this purpose, but recently was modified to also compute *p*-values for SNPs [63]. As this software relies on GEBV to calculate SNP effect and variance, blupf90 needs to be run first with three additional options in renf90.par:


OPTION saveGInverse

OPTION saveA22Inverse

OPTION snp_p_value


The first and second options save $G^{-1}$ and $A_{22}^{-1}$, respectively, and the third option saves $C^{u_{2}u_{2}}$, all in binary format. After running blupf90, renf90.par can be copied with another name, for example postgs.par, and the additional options are now:


OPTION readGInverse

OPTION saveA22Inverse

OPTION snp_p_value


The first two options are to read $G^{-1}$ and $A_{22}^{-1}$, respectively; the third option is used now to calculate *p*-values. A fourth option (i.e., OPTION windows_variance x) can be used if variance for SNP is to be calculated based on windows of x SNPs instead of individual SNP [58]. Based on Equation (48), postGSf90 needs GEBV, SNP content, and G^−1^ to compute SNP effect and variance. The first one is obtained from the blupf90 solutions file, the second from the SNP file, and the third from a file blupf90 created and named Gi. For the calculation of *p*-values, a file containing $C^{u_{2}u_{2}}$ was created by blupf90 and named xx_ija. After running postGSf90, several files are generated. One is snp_sol, which the column information is described in Box 4. 

Box 4Content of snp_sol file generated by postGSf90.1: Trait2: Effect3: SNP4: Chromosome5: Position6: SNP effect7: SNP variance8: Variance explained by n adjacent SNP (if OPTION windows_variance)9: Variance of the SNP solution (used to compute the *p*-value, if OPTION snp_p_value)

Three extra files are chrsnp, chrsnpvar, and chrsnp_pval, which are used to generate Manhattan plots for SNP effect, proportion of variance explained by *n* adjacent SNPs, and –log10 (*p*-value), respectively. Additionally, R and gnuplot scripts are also generated to create the Manhattan plots described above. Box 5 shows how to generate Manhattan plots in R and gnuplot. 

Box 5Creating Manhattan plots from files generated by postGSf90.For R users:  Rscript Sft1e2.R # Creates Manhattan plots for SNP effectRscript Vft1e2.R # Creates Manhattan plots for SNP varianceRscript Pft1e2.R # Creates Manhattan plots for SNP *p*-value For gnuplot users:gnuplot Sft1e2.gnuplot # Creates Manhattan plots for SNP effectgnuplot Vft1e2.gnuplot # Creates Manhattan plots for SNP variancegnuplot Pft1e2.gnuplot # Creates Manhattan plots for SNP *p*-value

The default formula to calculate variance or weight for SNP *i* is *d_i_ = 2p_i_ (1−p_i_)*
*a_i_^2^*. However, four different formulas are implemented in postGSf90 (http://nce.ads.uga.edu/wiki/doku.php?id=readme.pregsf90). A more robust way to compute SNP variance is the non-linearA shown in Equation (48). To change the SNP variance type to non-linearA, the following option should be added to the postGSf90 parameter file: OPTION which_weight nonlinearA

This option assumes the default constant (CT) is 1.125. To change the constant value to reflect a distribution closer to normal, use a CT value closer to 1:OPTION which_weight nonlinearA 1.05

By default, the maximum change in SNP variance is limited to 5, which is calculated as CT^(5−2)^ and returns a value of 1.4238 with CT = 1.125. If this limit is to be changed to 10, the following option can be used, where the value provided (x) is the result of the expression CT^(x−2)^. As an example, if CT is 1.05 and x is 10, the value provided to the option should be 1.4775:OPTION SNP_variance_limit 1.4775

A parameter file to run postGSf90 using non-linearA variance with CT equal to 1.05 and limit of 10, and computing *p*-value is in Box 6.

Box 6Parameter file for running postGSf90using non-linearA variance and computing *p*-value.
DATAFILE
 renf90.dat
NUMBER_OF_TRAITS
   1
NUMBER_OF_EFFECTS
   2
OBSERVATION(S)
 1
WEIGHT(S)

EFFECTS: POSITIONS_IN_DATAFILE NUMBER_OF_LEVELS

TYPE_OF_EFFECT[EFFECT NESTED]
 2  2 cross 3  12010 cross
RANDOM_RESIDUAL VALUES
  0.60000 RANDOM_GROUP  2 RANDOM_TYPE add_an_upginb FILErenadd02.ped
(CO)VARIANCES
  0.40000OPTION SNP_file genotypes.txtOPTION map_file gen_map.txt
OPTION readGInverse

OPTION readA22Inverse

OPTION snp_p_value
OPTION which_weight nonlinearA 1.05OPTION SNP_variance_limit 1.4775 

Although the calculation of SNP effect and variance was designed to be an iterative method, it is not recommended to use the iterative process when using the option to calculate *p*-value [63]. To check how to have weighted ssGBLUP where SNP effect, SNP variance, and GEBV are updated in an iterative way, see Appendix D.

### 2.9. Accounting for Sequence Variants in ssGBLUP

Genomic selection relies on linkage disequilibrium (LD) between SNPs and quantitative trait nucleotides (QTNs). By having dense SNP panels (i.e., >50,000 SNP), it is more likely that a QTN will be in LD with at least one SNP. If QTN A is linked to SNP B, depending on the strength of this linkage, once SNP B is observed it will imply QTN A was inherited together. Therefore, it is expected that increasing the number of SNPs the accuracy of genomic selection will increase. VanRaden et al. [66] showed an average increase of 1.6% in reliability of GEBV for a simulated trait when using 500,000 instead of 50,000 SNPs. According to Meuwissen et al. [67], the ideal SNP density is given by whole-genome sequence data. As millions of SNPs are screened, the causative variants are expected to be among them. However, the use of high-density SNP chips did not increase accuracy from medium-size chips, and use of sequence data is showing only marginally higher accuracies.

Using simulated data, Fragomeni et al. [68] showed that accuracy of GEBV in weighted ssGBLUP can approach 1 (i.e., perfect genomic prediction) if all causative variants are known and the true variance is assigned to each one of them. In a US Holstein dataset, Fragomeni et al. [61] tested the performance of ssGBLUP when using nearly 54,000 SNPs and when adding 17,000 significant variants discovered in a GWAS (pre-selected sequence SNPs) that involved 33 traits [69]. Although VanRaden et al. [69] reported an increase in reliability of GEBV of 4.3 points for stature by using non-linearA weights in a multistep scenario, no gain was observed in Fragomeni et al. [61] using either quadratic or non-linearA weight in ssGBLUP. This is possibly because the amount of data used in ssGBLUP overwhelms any a priori assumption made about SNP effects, making this method less sensitive to SNP weighting in the presence of large data. Another hypothesis to explain the steady reliability is that not all causative variants were present among the 17,000 significant SNPs. Although causative variants can be included in ssGBLUP assuming different weights for SNPs, maximizing the accuracy of GEBV would require the true identification of all causative variants, their substitution effect, their position, and the proportion of additive genetic variance they explain. To identify some of the causative variants, a large number of sequenced animals with phenotypes is needed. When a large amount of information is available, the accuracy may be high enough; therefore, improvements from the incorporation of causative variants are likely to be small for large data sets. When the number of genotyped animals is larger than the number of independent chromosome segments, the accuracy is maximized without SNP weighting/selection [70,71,72]. 

An optimal algorithm for finding causative SNPs would be to assign a large variance to those causative SNPs while reducing the variance of nearby SNPs or setting it to 0. This would resemble methods that perform a sequential estimation of SNP effects/variances. In such methods, SNPs close to causative variants are automatically disregarded. Although there are three different weighting methods implemented in BLUPF90 programs, any external weight can be considered, including the ones from Bayesian regressions. The only requirement is that those external weights have to be rescaled to sum to the number of SNPs used in the model. However, Gualdron-Duarte et al. [73] found that improvements in predictivity from unweighted GBLUP to BayesR and other methods were similar to improvements from unweighted ssGBLUP to weighted ssGBLUP with external weights. To check how to consider different weights or variance for causative variants in ssGBLUP, see Appendix D.

The default configuration of preGSf90, blupf90, and postGSf90 assume a maximum number of SNPs equal to 400,000. In the presence of sequence data (or over 400,000 SNPs), an extra option is required:OPTION maxsnp x

where x is the new maximum number of SNPs.

Although the most common way to include pre-selected sequence SNPs is to add them to the current SNP panel, it is possible to have an analysis where both are considered as separate components. In such a case, there is a need to fit two animal effects into the model—one for the current SNP panel and one for the pre-selected sequence SNPs [74]. However, the gains in accuracy using GBLUP with two animal effects was limited [74,75]. There are no reports on the literature about the use of SNPs from a panel and pre-selected from sequence fitting two H in ssGBLUP. Accommodating two G (GBLUP) or two H (ssGBLUP) is possible using BLUPF90 software, although this may have the same impact on accuracy as using different weights for pre-selected sequence SNPs. Additionally, a strong assumption of no correlation between the two random animal effects has to be assumed. 

To accommodate two genomic matrices in blupf90, the inverse of those two matrices should be constructed separately using preGSf90 and saved to a file. The preGSf90 has an option to save $H^{-1}$ but not H, because the latter is never constructed:OPTION saveHinv

This option saves the diagonals and upper diagonals of $H^{-1}$ as a plain text file (Hinv.txt) in the format of row, column, and value, where row and column are based on renumbered IDs. When using blupf90, the random type should be set as user_file (see RANDOM_TYPE in Appendix B) and the file name has to be provided (e.g., Hinv_chip.txt, Hinv_seq.txt, …). The random type user_file should be used as many times as the number of different SNP sets were used to compute $H^{-1}$(Box 7). 

For more details on how to use user_file, check the following link: http://nce.ads.uga.edu/wiki/doku.php?id=user_defined_files_for_covariances_of_random_effects.

Box 7Parameter file for running blupf90 using two **H**^−1^.
DATAFILE
 renf90.dat
NUMBER_OF_TRAITS
   1
NUMBER_OF_EFFECTS
   3
OBSERVATION(S)
 1
WEIGHT(S)

EFFECTS: POSITIONS_IN_DATAFILE NUMBER_OF_LEVELS

TYPE_OF_EFFECT[EFFECT NESTED]
 2   2 cross 3   12010 cross 3   12010 cross
RANDOM_RESIDUAL VALUES
  0.60000 RANDOM_GROUP  2 RANDOM_TYPE user_file FILEHinv_chip.txt
(CO)VARIANCES
  0.30000
RANDOM_GROUP
 3 RANDOM_TYPE user_file FILEHinv_seq.txt
(CO)VARIANCES
  0.10000 

### 2.10. Large-Scale Genomic Evaluations with ssGBLUP

The most expensive operation in ssGBLUP, as implemented in Aguilar et al. [16] and Christensen et al. [17], is the inversion of G and $A_{22}$. This operation has an approximately cubic cost with the number of genotyped animals. With efficient computing algorithms, matrix inversion is feasible for up to 100,000 genotyped animals [76,77]. The number of genotyped animals in some livestock species goes far beyond 100,000 and considerably increases every year. One example is the American Angus Association that has over 780,000 (Steve Miller, 2020; personal communication) and the US dairy industry has already collected over 3.4 M Holstein genotypes (https://queries.uscdcb.com/Genotype/counts.html), where only 11% of those are for males, over 75% are for animals without a BLUP evaluation, and there is a very slow increase in the number of genotypes for proven bulls [78]. 

To overcome the limitation set by the number of genotyped animals in ssGBLUP, Misztal et al. [79] proposed the algorithm for proven and young (APY) to construct $G^{-1}$ without having to explicitly invert G. The APY is based on the principles discovered by Henderson [50] and Quaas [51] to recursively construct the inverse of A. The logic behind the construction of $G_{\mathrm{APY}}^{-1}$ is that the genotyped animals are split into core (*c*) and noncore (*n*), and the main assumption is that breeding values for noncore animals ($u_{n}$) are functions of breeding values of core animals ($u_{c}$):(52)$$u_{n}=P_{nc}u_{c}+\Psi_{n}\;$$
where $P_{nc}$ is a matrix that relates breeding values for noncore to core animals, and $\Psi_{n}$ is a diagonal matrix with estimation errors. Following further developments [80], $G_{\mathrm{APY}}^{-1}$ can be constructed as:(53)$$G_{\mathrm{APY}}^{-1}=\left[\begin{array}{cc} G_{cc}^{-1} & 0\\ 0 & 0 \end{array}\right]+\left[\begin{array}{c} {-G}_{cc}^{-1}G_{cn}\\ I \end{array}\right]M_{nn}^{-1}\left[\begin{array}{cc} {-G}_{nc}G_{cc}^{-1} & I \end{array}\right]$$
with $m_{nnn_{ii}}=g_{ii}-{\;g}_{ic}G_{cc}^{-1}g_{ci}$. The APY algorithm creates a generalized sparse inverse of G at approximately a linear cost in computing and storage [79,80] and has been extensively tested for beef cattle [18], dairy cattle [81,82], and pigs [83,84]. This algorithm enables ssGBLUP evaluations with millions of genotyped animals, as the only inverse needed is for the core animals. Pocrnic et al. [85] and Pocrnic et al. [86] found that the ideal number of core animals depends on the dimensionality of genomic information. Even though millions of animals can be genotyped, the amount of independent genomic information or independent chromosome segments is limited and depends on the effective population size (Ne) and genome length. The knowledge about this non-redundant information enables computations with large-scale genomic data. Pocrnic et al. [86] found that the minimum number of core animals was around 4000 for pigs and chicken, 11,000 for Angus, 12,000 for Jerseys, and 14,000 for Holsteins. 

If $G_{\mathrm{APY}}^{-1}$ is efficiently computed but $A_{22}^{-1}$ is not, ssGBLUP cannot be used for over 100,000 genotyped animals. To avoid explicit inversion of $A_{22}$, Stranden et al. [87] and Masuda et al. [88] proposed to compute an efficient inverse indirectly as a product of sparse matrices:(54)$$A_{22}^{-1}=A^{22}-A^{21}{\left(A^{11}\right)}^{-1}A^{12}$$
where $A^{11}$, $A^{21}$, and $A^{22}$ are portions of $A^{-1}$ for non-genotyped, between genotyped and non-genotyped, and for genotyped animals, respectively. Computing time for constructing $A_{22}^{-1}$ for 570,000 genotyped animals extracted from a population of 10 M animals was around 11 min [88]. Single-step GBLUP with $G_{\mathrm{APY}}^{-1}$ and efficient $A_{22}^{-1}$ was successfully applied to over 10.9 M cows with milking records, 13.5M animals in the pedigree, and about 2.3 M genotyped Holsteins [89]; using 15,000 core animals, the complete evaluation for a model with 18 type traits took four and a half days to converge. Within this time stamp, the construction of $G_{\mathrm{APY}}^{-1}$ and $A_{22}^{-1}$ took one day. In fact, this time depends on the total number of genotyped animals and the core group size.

Unfortunately, the subroutines to create $G_{\mathrm{APY}}^{-1}$ and efficient $A_{22}^{-1}$ are not implemented in the free distribution of BLUPF90 family of programs.

### 2.11. Unknown Parent Groups (UPG) and Metafounders in ssGBLUP

Commercial populations, especially sheep, beef and dairy cattle, often have incomplete pedigrees. In BLUP, missing parents are modeled by UPG [51,90,91]. Such groups are also known as phantom parents or genetic groups and are used to represent the average level of breeding value in a group where parents were missing. Different groups can be assigned based on year of birth, sex, breed combination, etc. As UPG are mainly modeled as fixed effects, they need to be defined carefully to be estimated accurately and to avoid confounding with other effects in the model [51]. In ssGBLUP, when UPG are applied only to pedigree relationships, the convergence rate can be slow [92]. Misztal et al. [93] revised UPG equations to include groups also in the genomic portion of $H^{-1}$, which then becomes $H^{-1*}$, based on Quaas–Pollak (QP) transformation [90]:(55)$$H^{-1*}=A^{-1*}+\left[\begin{array}{ccc} 0 & 0 & 0\\ 0 & G^{-1}-A_{22}^{-1} & -\left(G^{-1}-A_{22}^{-1}\right)Q_{2}\\ 0 & -Q_{2}^{\prime }\left(G^{-1}-A_{22}^{-1}\right) & Q_{2}^{\prime }\left(G^{-1}-A_{22}^{-1}\right)Q_{2} \end{array}\right]$$
where $Q_{2}$ is a matrix that relates genotyped animals to groups; $G^{-1}$ can be replaced by $G_{\mathrm{APY}}^{-1}$ in large genotyped populations. When UPGs were applied to all components of $H^{-1}$, convergence dramatically improved for a multiple-trait model in the Nordic dairy cattle population [94]. Revised UPGs also worked well for the US Holstein data up to 2014 [56]. However, using data updated to 2015, Masuda et al. [95] reported lower prediction reliabilities using revised UPG than not using UPG at all. Therefore, it is not clear whether ssGBLUP equations should include UPG for $G^{-1}$, as genomic relationships are not affected by missing pedigree, implying UPG are automatically accounted for. Tsuruta et al. [96] showed that UPG for $G^{-1}$ were not estimable for young genotyped animals in an 18-type trait genomic evaluation.

Current use of UPG in BLUP ignores the fact they represent sets of related, missing parents in a population under constant selection. Thus, a more accurate modelling would assume missing parents can be related and inbred [97,98]. Legarra et al. [99] proposed the idea of metafounders, which are “inbred and related” UPG. In ssGBLUP, the genomic relationships are usually derived based on current allele frequencies and scaled for compatibility with pedigree relationships as in Vitezica et al. [48]. Based on the metafounders theory, G would be derived using 0.5 allele frequencies as an “absolute reference” [100], and A would be scaled for compatibility with G using covariances among and within metafounders. According to Legarra et al. [99] the covariances represent size of the base population at the time when pedigree recording started and they would be estimated in such a way so that they account for scaling, unaccounted inbreeding, and different genetic level (i.e., when using multibreed or selected populations). Several methods were proposed to estimate the covariances among metafounders, including via gene frequencies related to unknown parents [101]. In simulations and real data, the concept of metafounders delivered the least biased predictions [38,101]. In ssGBLUP, $H^{-1}$ with metafounders ($H^{\Gamma -1}$) can be represented by:(56)$$H^{\Gamma -1}{=A}^{\Gamma -1}+\left[\begin{array}{cc} 0 & 0\\ 0 & G^{-1}{-\;A}_{22}^{\Gamma -1} \end{array}\right]$$
where **Γ** is the relationship matrix among metafounders, which can be estimated using generalized least squares [101]. Once **Γ** is inverted, Henderson [50] rules can be used to construct the inverse of the pedigree relationship matrix. As metafounders are based on UPG, estimating **Γ** strongly depends on how the groups are assigned. Estimating **Γ** is an active area of research as many UPGs are weakly related to genotyped individuals.

In the BLUPF90 family of programs, renumf90 can create UPG based on year of birth or can recognize negative values in the pedigree as UPG (see Appendix B). If blupf90 is used to run ssGBLUP, UPG will be set only for $A^{-1}$. To set UPG for the full $H^{-1}\;$ like demonstrated in (54), the following extra option is needed: OPTION exact_upg

To set UPG only for $A^{-1}$ and $A_{22}^{-1}$, a second option is needed to remove UPG for $G^{-1}$: OPTION TauOmegaQ2 0.0 1.0

As the metafounders concept is still recent, the BLUPF90 developers are currently working on a standalone software to estimate **Γ**, which is called gammaf90. After that, blupf90 will be modified to accept an extra type of random effect specific for metafounders. Independent software used in Garcia-Baccino et al. [101] to estimate **Γ** and to compute $H^{\Gamma -1}$ with instructions and examples can be found here https://github.com/alegarra/metafounders. After $H^{\Gamma -1}$ is constructed, blupf90 can be used with the random type set as user_file (see RANDOM_TYPE in Appendix B). 

## 3. Conclusions

The BLUPF90 is a complete software suite for the most common computations needed in animal breeding and genetics. The programs are highly optimized and have been under constant development for over 20 years. Single-step GBLUP, which is one of the main tools for genomic analysis, was first implemented in 2009. Since then, several changes were made to make ssGBLUP flexible to any model, number of traits, number of phenotypes, number of genotyped animals, and sequence data. Single-step GBLUP is fully supported in the BLUPF90 software suite and has been used for genomic evaluations worldwide.

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
