# Peer review of "Single-Step Genomic Evaluations from Theory to Practice: Using SNP Chips and Sequence Data in BLUPF90"

_genes, 2020, doi:10.3390/genes11070790_

Round 1
Reviewer 1 Report
The most important contribution of this paper is that it provides an easy-to-follow, in-depth explanation on how ssGBLUP works, while clearly stating its differences with other methods. This will be of the interest of many users and will help to avoid confusion, and worth publication, so I only have a few minor comments.
The article is centered on the blupf90 suite, and the main issue I found is thinking on potential new users. The article structure with sections explaining/discussing theory and practice is adecquate, but the first practical steps provided in the article should be simplified. These steps are included in three different sections (2.4, appendix A and B), and not always sorted or fully explained. Eg. A is mentioned early in the article to download and "use" the software, but data and parameter files are not introduced until 2.4 and actually B as to be done in advance. Reorganize this in a more straightforward way. Alternatively, a combined appendix with all the required steps and commands in order will be of great utility (like a pipeline). Hopefully this will attract new users, and broad the audience of the article.
Other minor comments/suggestions:
- ssBR methods are only little discussed in the article, I'd suggest to remove its mention from the abstract.
- Cite other software that does run BLUP (eg. GCTA or R packages) and blupf90 advantages.
- Appendix B contains too much options, not all of them used. Wonder if this section could be more focused.
- Since a github repo is provided with data example, it should include as well parameter files as required to reproduce examples in this paper, so users do not need to copy them from it.
- Review the different binaries provided. I can tell that at some point there was a blupf90 with a number version higher than 1.58 (1.68?) that returned segfault, and that OPTION spn_p_value is only available for postGSf90 binaries 1.64 (eg. in the link for Linux 64) and not earlier releases (as. 1.45 for the 32 bit version). In any case, v1.64 returns also a segfault for me. I am aware that falls out of the scope of the article and the authors might not be able to change it, but if that were the case, please make versions consistent, and provide examples easier to reproduce, or with meaningful message errors. Open source would greatly help solving these issues.
- line 52: EBV should be defined
- Sentence 62-66: provide some references.
- line 760: numer->number
Reviewer 2 Report
A very good review which helps to understand the developments and application of the ssGBLUP and BLUPF90 software suite. The authors also provide clear instruction for researchers to access and use the software for various needs.
I will however suggest that the authors should as much as possible link the some of the equations with the text. Two examples are provided below please:
i. Edit Lines 146 and 147 to read "If we express the breeding values of the animals i and j as ma (with m a row vector) deviated from the population mean [27], we obtain equations (2) and (3):".
ii. Edit Lines 192-193 to read: "Using a simple animal model as shown in equation (14):"
